# Equine-Assisted Therapeutic Intervention in Institutionalized Children: Case Studies

**DOI:** 10.3390/ijerph20042846

**Published:** 2023-02-06

**Authors:** Ana Rita Matias, Graça Duarte Santos, Nicole Almeida

**Affiliations:** 1Departamento de Desporto e Saúde, Escola de Saúde e Desenvolvimento Humano, Universidade de Évora, 7004-516 Évora, Portugal; 2Comprehensive Health Research Centre (CHRC), Universidade de Évora, 7000 Évora, Portugal; 3Psychomotor Therapist at T-Terapias, 2500 Caldas da Rainha, Portugal

**Keywords:** equine-assisted services, socioemotional competence, psychomotor intervention

## Abstract

A significant number of institutionalized children have behavior problems. Socio-emotional skills are fundamental for their adaptation and success throughout life and are usually weakened in this population. Equine-assisted services (EAS) are a form of therapeutic mediation that facilitates and requires the practitioner’s participation, contributing to the promotion of various psychomotor and socio-emotional dimensions. This study was developed during 17 sessions of EAS with a psychomotor intervention, which took place individually and weekly and lasted approximately 45 min, with three institutionalized children. A quantitative and qualitative assessment was carried out before and after the intervention to study the effects of an EAS intervention on socio-emotional competencies in the three institutionalized children. There was an improvement in skills, with an impact on intrapersonal skills and marked improvement in self-regulation and self-control, in addition to an improvement in the intentionality of movement and adequacy of gesture to the context. This type of intervention underlies a renewed educational and therapeutic approach, contributing to mental health promotion in this population.

## 1. Introduction

A child’s development depends on a combination of internal and external factors [1]. According to Strech [2], a child’s life can have multiple evolutions. Depending on what is provided to them throughout childhood, the child may present diversity in their way of feeling, thinking, and acting. It is generally with the family that most children develop in an ideally healthy environment. However, these conditions, which are conducive to success in child development, are only sometimes ensured, thus compromising the child’s safety, well-being, learning, and development [2,3,4].

Institutions and temporary reception centers are one of the answers adopted if the child or young person is at risk. These sites are typically characterized by the absence of regular individualized care and low levels of caregiver–child social interaction due to team turnover and the reduced number of caregivers in these institutions [5].

When children or young people are institutionalized, it is essential to understand the impact that the separation process can have on their psychological life as well as the disturbances that can result from this painful process because feelings of guilt and responsibility are common, giving rise to feelings of loss of caregivers [6,7]. In this sense, this event leads to the loss of affective and spatial references, as the child experiences confrontation with new places, routines, and unknown people [6].

About 68.6% of the 6706 institutionalized children and young people have distinct psychological characteristics, principally the presence of behavior problems, which corresponds to 27% of this figure, 13% of the children and young people are in a situation of clinically diagnosed mental weakness or disability, and 4.5% have clinically diagnosed mental health problems [3]. Regarding the level of follow-up/support for mental health, about 35% of in the institutions are monitored regularly in psychology and 25% receive regular monitoring in psychiatry [3].

So often weakened in this population, socio-emotional skills are a multidimensional construct, fundamental for humans, particularly for their adaptation and success throughout life. Among the competencies associated with the domain of intrapersonal skills are that they make it possible to establish realistic goals, to succeed in emotional regulation and self-control, and to develop positive thoughts and coping strategies, among others [8]. These competencies also relate to the mastery of interpersonal skills essential to the interactions that allow listening and communication skills, the analysis of different perspectives, success in negotiation, and problem-solving skills [8].

The social development of children is strongly influenced by their peers, particularly by the relationships that occur in the context of peer groups [9]. Young children recognize and understand the differences in the quality of their friendships based on aspects such as validation, aid, exclusivity, and levels of conflict [10]. Friendship processes are strongly associated with levels of satisfaction and stability, i.e., children are more satisfied with their friendships when they offer higher levels of validation and exclusivity and lower levels of conflict and are more likely to maintain them over time. Children with high levels of conflict are associated with difficulties in the school adjustment process, being influenced by more aggressive interaction styles and misfit behaviors, among others. Friendships that offer higher levels of ego validation, self-affirmation, interpersonal help, and the resolution of social problems tend to increase the feeling of competence and security [10].

Children who have experienced poor interactions with their peer group have a higher risk of social maladjustment [11], emotional [12], academic [13], and social adaptation problems in adulthood [14].

Considering renewed educational and therapeutic approaches that promote these populations’ adherence to developing their socio-emotional skills is a constant challenge.

Psychomotor intervention has been included in one of the three significant areas of intervention in an equestrian context, corresponding to the areas of therapy, learning, and education, with prominence in the first aspect. The work developed is directly related to the area of training, incorporating horses in the intervention, and recognizing their potential to achieve the therapeutic objectives [15].

Animal-assisted therapy (AAT) refers to the interconnection of animals with people in a health context. It is directed by professionals (such as psychomotor therapists) with knowledge in the health area and who consider the individuality of the person, and it aims at promoting physical, social, and cognitive well-being [16]. The dimension of empathy, imperative for healthy emotional and social functioning, is one of the most focused on; intrinsically linked to early parental experiences, it is often weakened in this population [17,18].

Within AAT, we highlight equine-assisted services (EAS). This form of therapeutic mediation facilitates and requires the participation of the practitioner, contributing to the promotion of various psychomotor and socio-emotional dimensions of the subject, such as a greater capacity for relaxation and awareness of their body, an increase in balance and coordination (with the social advantages arising from it), joint mobilization and strengthening, and enhanced self-regulation of affections and aggressiveness. The interaction with the horse also promotes the development of new forms of socialization, self-confidence, and self-image [19].

Burgon [17] developed research relating to EAS and children at risk, verifying that this population developed significant relationships with horses. It showed that children with marked socio-emotional problems established affective connections with horses in different ways and, in many cases, showed empathy for animals.

In addition to the benefits that this environment brings to the social integration of these children and young people, EAS leads to significant emotional and psychological benefits [20]. Caqueret [20] states that a therapeutic the physical touch in the relationship with the horse promotes several elements, namely the mobilization of the body, strengthening, confidence, aggressiveness control, and the development of affections.

For this investigation, observation and listening capacities were essential and are highlighted by Rodriguez et al. [21] as dimensions of the psychomotor therapist profile. Observing is a fundamental capacity that precedes any intervention. When observation is performed, the simultaneous realization of listening is implicit; that is, listening ensues when an observation occurs, and both imply an understanding of what is expressed through the bodily motions of the child. Also implied is the ability to empathize, trying to understand how a person is feeling and what are the meaning/meanings of their expressions, and consequently offering an adequate response from an empathic and affective relationship [21].

This research, presented in this paper, aimed to understand the effects of an EAS intervention on socio-emotional competencies in three institutionalized children.

## 2. Materials and Methods

### 2.1. Participants

The participants were selected by convenience and included five children, aged 5, 6, and 9 years, institutionalized in a temporary reception house (CAT). The names used in this study are fictitious, and the participants did not have a diagnosis. Three of the five children who began the intervention met the criteria for the research, so only complete data on eligible children are presented. The criteria for inclusion in the study were: (1) being institutionalized in a temporary reception home; (2) participation in the intervention program with greater than 80% attendance and; (3) the legal guardians and teachers of the children being available to answer the proposed questionnaires.

### 2.2. Ethics Approval

The university research ethics committee approved all the procedures. In addition, participants gave written informed consent, according to the Helsinki Declaration (approval number: 15007), before participating in this study. All children gave their verbal consent.

### 2.3. Procedures

After the initial clarification and the signing of the Informed Consent Requests, all tutors and participants had the opportunity to ask any questions and clarify all doubts regarding the investigation and respective methodological steps. After the definition of the study participants, anamnesis data and other general information about the children were collected by the therapist before the study began.

Twenty sessions were planned: two for the initial assessment, seventeen for the intervention, and one for the final evaluation. Considering the importance of the first contact with the children, two sessions were carried out in the institution. The contact with the horse allowed for the establishment of a therapeutic relationship between the children and the therapist. Considering the overall objectives of the study with these children, the data collected by the questionnaires and observational data, therapeutic goals, and respective therapeutic projects were designed based on the individual characteristics of each child, as well as a general intervention guide seeking to encompass global needs. Then, 17 sessions of EAS were developed (psychomotor intervention), taking place individually, on a weekly basis, lasting approximately 45 min, and in a partner institution. During these sessions, five moments of observational data collection in an equestrian context were performed (in the 1st, 5th, 9th, 13th, and 17th sessions), which helped to organize the therapeutic progression. Finally, after the 17th session, a final evaluation was performed using the same instruments, and proceeded to compare and interpret the results obtained.

### 2.4. Instruments

#### 2.4.1. Strengths and Difficulties Questionnaire (SDQ)

The SDQ is a brief behavioral screening questionnaire consisting of 25 items, that allows for the assessment of the abilities and difficulties of children and young people aged between 4 and 17 years [22]. It provides information about the behaviors of the child or young person over the last six months in their most commonly used environments, such as school or home [23]. The 25 items of the SDQ are distributed in five scales (each with five items) evaluating conduct problems, hyperactivity–inattention, emotional symptoms, peer problems, and prosocial behavior. The total score of each subscale varies between 0 and 10 points, with three alternatives for ranking the statements: “not true” (0 points), “it’s a little true” (1 point), and “it’s very true” (2 points). Based on the answers, a score is obtained for each subscale evaluated, as well as a global difficulty index that corresponds to the sum of all subscales, except the prosocial behavior scale [23]. This questionnaire defines cutoff values that classify the child’s behavior according to low risk, borderline (moderate risk), or atypical (high risk). In the total scoring scale of difficulties, a score of 12 points is defined as being situated on the borderline, and a score above 16 points is atypical. The second scale of this questionnaire has a borderline value of 5 points, and from 6 points is classified as atypical. The scale of behavior and hyperactivity problems is defined with borderline values of 3 and 6, and values of 4 and 7 correspond to atypical, respectively. The scale of relationship problems with colleagues indicates cutoff values of 4 and 5, the first being within the borderline classification and the second within atypical. Finally, the scale of pro-social behaviors is defined with a classification of 5 for borderline and a value of 0 for atypical.

#### 2.4.2. Assessment of Children’s Emotion Scales (ACES)

ACES [24] is an instrument consisting of three subscales, applied in the context of an interview with the child. It aims to evaluate the children’s ability to recognize emotions and the accuracy of their attribution. In this investigation, the adapted version for the Portuguese population was used. It consists of three subscales: facial expressions (20 items that assess the ability to understand the emotions transmitted by the facial expressions of other children), emotional situations (15 items that examine the child’s ability to recognize emotional arousal in social situations), and behaviors (15 items). The scale evaluates the emotional knowledge of the child through the ability that they demonstrate to correctly associate with each stimulus (facial expression, situation, or behavior) one of five possible emotions (happy, sad, angry, frightened, and normal) [25]. For the latter emotion, the child is asked to identify the emotion presented by another child among five possible response alternatives: contented, sad, angry, frightened, and, finally, normal. While the first four emotions are evident, the fifth alternative is ambiguous and, therefore, does not refer to any specific feeling. The result of this question on the three scales is presented as the correct emotional perception (PEC) [25].

#### 2.4.3. Individual Psychomotor Observation Sheet

The individual psychomotor observation sheet was developed from the individual observation model described by Martinez et al. [26]. This model allowed us to understand motor expressions and gestuality, highlighting how a child uses their body, their motor efficiency, how the child guides, how they relate to objects and space, and others [27].

With this form, we observed movement (quality of movement, balance, coordination, speed, rhythm, lateral dominance, hand–eye coordination, and body mobility), gestuality (facial, corporal, and empathic expression), posture (postural tone, postural preferences, and postural adaptation), tone (basic, movement, and overall motricity), space (exploration, orientation and structuring, intra- and extra-personal space), time (time adjustment, individual and shared rhythm, time of use of space), objects (qualities, use, respect for others, meaning), relationship with others (manifestation of needs, attitude towards the game, mode of communication), and representation (graphic, construction or modeling).

The response to each parameter was made according to what was observed in each session. In the first session, each parameter had three possible responses: “adequate”, “unsuitable”, or “no register”. In the following sessions, for this answer, there were 3 possibilities for recording “+” (if there was an evolution in the previous session), “−” (if there was involution), and “=” (if maintained). Ahead of each answer, there was a space for registration in the qualitative evaluation text.

This observation model was adapted to the equestrian context and applied throughout the sessions, contemplating the moments on horseback and the interaction with the horse (Appendix A).

### 2.5. Intervention

The intervention, in general terms, was delineated with the common objectives of favoring the affective relationship (for reduction in emotional symptoms), promoting relaxation, adhering to rules, solving problems with identification and/or development of appropriate solutions (to reduce hyperactivity and behavior problems), and developing the relationship with the horse (intending the development of empathy).

The dynamics of the sessions were based on a psychomotor perspective with semi-directed and individualized sessions, adapted to each case’s individual rhythm, needs, and capacities. Moreover, a phased methodology was used to promote the socio-emotional objectives mentioned. At the beginning of each session, activities were prepared between the psychomotor therapist (the only person present) and the child, i.e., the child could choose to remain in the covered riding arena and carry out interaction activities with the horse, ride a horse, carry out this interaction abroad (for example, walking the horse with a guide), or carry out recreational games (or activities of plastic expression), considering the materials available in the riding arena (always with the involvement of the horse) and also in accordance with the needs/availability of each child. At the end of each session, a moment of reflection was made so the child could express (through drawing or verbalization) what the session’s most pleasant moment was. The mounted and unmounted work was variable, depending on the children’s motivations.

The horse used was 5 years old, had been trained for the two years prior to the study, and had been used only for therapeutic interventions in the year before this study. It is a calm and gentle horse.

Psychomotor perspectives focus on motor actions; in this sense, movement, gestuality, and the way the child relates to time, space, objects, and others are continuously represented in maps of the brain, and what exists in the mind is instantly disseminated in the body and the actions of the child in relation to the environment. This perspective allows us to link gestural communication to verbal communication and language, that is, our internal experience of the other’s actions, emotions, and intentions [28].

At an early stage, the focus was on acquiring a greater body awareness and identifying body positions and states, as well as the children’s association with their own emotions. Next, the intention was to associate the strategies of balance, locomotion, tonicity, etc., with the identification, recognition, understanding, and communication of basic emotions (such as joy, sadness, anger, and fear) in the children themselves and others (the horse), promoting awareness of the relationship (with the horse and with the therapist) through the understanding of their own behavior and taking the others’ into perspective.

## 3. Case Description

### 3.1. Case 1

Maria was 9 years old at the time of the study and is the sixth daughter of a group of eight siblings, (two in the same institution as Maria and a third in another institution). At the time, her 34-year-old, unemployed mother lived with a man (36 years old and unemployed) who represented the father figure. It is not known who her biological father is. Maria was institutionalized in July 2021 due to family neglect, which had an impact on her health and education. Her family’s context was not beneficial to her health, and she usually did not have adult supervision. The presence of the police in the house was frequent due to a history of domestic violence and minor crimes (robberies) by the older brothers. Maria’s medical history refers to changes in her level of diction and frequent headaches. The integration into the host house was without complications on the part of the child, with a good adaptation to the routines and norms of the institution observed. The head of education (social worker of the institution) stated that in family visits, no affinity/proximity to her mother was displayed, and Maria was not bothered by her departure. At the time of the study, the child continued to keep in touch with family members through face-to-face visits and phone calls. Without ever having attended preschool, Maria was inserted in the fourth year of schooling, under the inclusive education regime with an adapted curriculum. From an emotional and relational point of view, she presented herself as a child who often seemed unhappy, requesting a lot of attention from the adult and revealing affective needs. With impulsive behaviors, she easily self-regulated and presented tantrums that were difficult to contain. She related easily to colleagues, although she preferred to play with younger children or adults over peers, with whom she sometimes had aggressive interactions. According to the head of education, these behaviors were only to call for attention. In playful terms, she showed a preference for engaging in symbolic play with peers, activities of care for others, especially the babies of the institution (mothering behaviors), and painting and drawing activities.

### 3.2. Case 2

Tomás was 5 years old at the time of the study and belongs to a family with three brothers. The mother had been unemployed for several years prior to the study, surviving through subsidies and outside aid. A stepfather represented the child’s father figure; his biological father was unknown. Institutionalization occurred due to poor living conditions, a history of domestic violence, living without the supervision of an adult, and neglect in health and education. His medical history suggested the development of insomnia. The child’s integration in the host house in 2020 was without complications, despite evidence of some unadjusted behaviors such as periods of absence and difficulty in controlling the sphincter, with the observation of improvements eight months after the reception. He attended preschool at the time of the study. He was described as a sweet and happy child, despite sometimes being impulsive, having difficulties with emotional self-regulation, and losing control quickly. He showed difficulty in expressing his needs and feelings, addressing an adult only in a conflict situation to make complaints. In playful terms, some of his favorite games were symbolic play, namely, playing “families”, building houses, playing with objects that represented cartoons, and games that required broad movements or cognitive mastery activities (puzzles, Lego). At the time of the study, Tomás was attending swimming lessons to adapt to the aquatic environment.

### 3.3. Case 3

Nuno was 6 years old at the time of the study and had eight brothers (two of whom are in the same institution, Daniel and Maria). Their 34-year-old mother, at the time of the study, was unemployed and lived with a man (36 years old and unemployed) who represented the father figure. It is not known who the biological father is. Nuno was institutionalized in July 2021 due to family neglect with health and education consequences. The family housing did not have hygienic or safe conditions, and Nuno was usually unsupervised by an adult. The presence of the police in the dwelling was frequent due to the history of domestic violence and records of minor crimes (thefts) by the older brothers. His medical history suggests there were changes in the level of speech, particularly in the joint, which were followed up in weekly speech therapy. His integration into the host house took place without complications, and he was observed to have a good adaptation to the routines and norms of the institution. The head of education mentions little affinity and proximity in the mother’s visits, and that he was not bothered by her departure. At the time of the study, he was attending preschool, having only been inserted in 2022. Emotionally and relationally, it was easy for him to form new friendships, although he preferred to play alone. When faced with less adjusted behavior from colleagues, he was shown to be dissatisfied.

## 4. Results

### 4.1. Case 1

#### Initial and Final Evaluation

The results of the SDQ are presented below (Figure 1), derived from the responses of parents and teachers collected in two moments, before and after the intervention. Figure 1 represents the overall values obtained, allowing verification of the evolution at the first moment of evaluation of the person in charge of education and a teacher.

Figure 1 shows the impact of the intervention on Maria’s competencies and difficulties from the respondents’ perspective. It simultaneously presents the evolution of the total subscales in both moments as well as the total score of the difficulties, which corresponds to the sum of all subscales except that of prosocial behaviors. The initial evaluation of this screening instrument indicated a high risk of emotional and behavioral difficulties, revealing an atypical classification in all scales except that of prosocial behaviors. In the post-intervention evaluation, the results reveal a low risk of emotional and behavioral difficulties in all subscales and a maximum score in the subscale of prosocial behaviors. In conclusion, there are very significant differences between the two evaluation moments.

The results of the ACES are presented below. The first three graphs show the values of each subscale in the pre- and post-intervention moments corresponding to the subscales of facial expressions (Figure 2a), situations (Figure 2b), and behaviors (Figure 2c).

Figure 2a reflects difficulties in recognizing facial expressions of “sadness”. At the first moment of evaluation, there was an incorrect identification of some emotions, with “normal” misinterpreted as the negative emotions “scared” or “angry”. The emotion that was more consistently recognized correctly was “frightened”. After the intervention, there was a significant improvement in the identification of facial expressions corresponding to contented, angry, and normal, maintaining the difficulty in identifying the facial expression of sadness.

Figure 2b, referring to the data from the situation’s subscale, shows evidence of the initial difficulty in correctly associating situations described with emotions, especially those that could have the classification “angry”, “scared”, or “normal” (without any correct answer). In addition, there was not only little emotional discrimination but also a specific emotional interpretation. Of the fifteen items in the subscale, nine were associated by Maria with “sad” emotions. In the final evaluation, significant improvements were observed after the intervention, obtaining maximum scores in all situations except those that should have been classified as “normal”. Neither in the initial nor in the final evaluation was there any correct recognition of a “normal” situation.

In the data from the behavior subscale (Figure 2c), Maria showed difficulty associating emotions with phrases representative of their behaviors, especially those that corresponded to “scared” or “normal”. Given the detailed analysis, it is possible to observe some confusion in the attribution of emotions to behaviors; for example, in the face of a behavior in which the child would be expressing herself “as angry”, Maria classified her as “happy” and in other situations whose correct classification would be “normal”, Maria classified them as “sad” and “angry”. In the final evaluation, the results improved for all items (with a maximum score of 3) except for those that should have been cataloged as usual (which still need to be identified).

PEC values can be seen in Figure 3.

Figure 3 provides an overall analysis of Maria’s performance, with significant changes in all subscales between the two evaluation moments. The positive evolution in the different subscales is reflected in the level of correct emotional perception, which progressed from 22 to 36 points in the evaluation after the intervention.

The data collected in five moments through the individual psychomotor observation sheet allowed a look at Maria’s therapeutic progression. Regarding movement, Maria revealed some instability in progress throughout the sessions. However, she showed a tendency to increase body mobility in a more controlled way. The same trend was verified in gestures, in which Maria revealed greater disinhibition and a body and empathic expression more adjusted to the situations. There were no differences in posture and tonicity that were initially adequate. In the relationship with space, there was progressively greater exploration of space and a smaller invasion of the space of others, showing greater awareness of their personal space. The relationship with time was evidenced by some instability, revealing a tendency towards a greater capacity to adjust to the individual rhythm and the shared rhythm (with the horse). Regarding the relationship with others (the horse and therapist), Maria improved in the manifestation of needs (verbally and non-verbally), with an attitude of greater cooperation towards games. She showed an initial tactile relationship with the horse, touching with her whole hand, and, at a later stage, a body relationship in which she touched the horse’s entire body while walking and performing the care activities. Finally, the representations of Maria, which initially were not very detailed, began to become more invested and organized and directed toward the horse.

### 4.2. Case 2

#### Initial and Final Evaluation

The results of the SDQ from the parent and teachers are illustrated. Figure 4 represents the overall values obtained in two moments of evaluation, before and after the intervention.

The impact of the intervention on Tomas’ skills and difficulties from the perspective of respondents. At the level of the subscale of problems with emotional expression, the values between the moments of evaluation were identical, contrary to what occurred in the school context, in which there was a significant improvement in all parameters of the subscale. In the hyperactivity subscale, there was a tendency for positive evolution in both contexts. Regarding the subscale of relationship problems with colleagues, the guardian did not identify changes between the moments of evaluation, contrary to what was verified in the school context, in which the teacher indicated that Tomás began to relate better with others. Regarding the subscale of prosocial behaviors, in the domestic context, a decrease in sensitivity towards others was identified, contrary to what occurred in the school context, which showed an improvement in all points.

The results of the ACES are presented (Figure 5).

Figure 5a reflects difficulties in recognizing facial expressions of “sadness” and “normal”, expressed in the first moment of evaluation. After the intervention, there were improvements in the recognition of the facial expression “normal” compared to the previous comment, Figure 5b, referring to the data of the situation’s subscale, highlights the ability to correctly associate situations described with emotions, especially those that would have a classification “sad” and “content” in which it obtained maximum classification. Figure 5c (behavior subscale) indicates that at the first moment of evaluation, it misidentified some emotions, being “normal” or “frightened” misinterpreted as “sad” or “angry”. After the intervention, there was a significant improvement in identifying emotions present in “scared” and “normal” behaviors.

Figure 6 allows an overview of Tomas’ performance, reflecting a positive evolution in correct emotional perception.

The data collected from the psychomotor observation sheet allowed us to observe that throughout the sessions, Tomás proved to be constant at the level of movement, presenting identical qualities in terms of equilibrium, coordination, speed, and rhythm. In terms of gestures, he progressed from an inadequate and sometimes exaggerated expression to a greater ability to adapt his expression. Tomás improved his empathic expression toward the horse, showing himself to be more conscious and empathetic toward him. There were no changes in tone, which was already adequate. However, there was an improvement in the level of horse riding. In the relationship with space, there was a progressively greater exploration of the space, alternating initially between the space close to the therapist and the horse or at the top of the ramp, and a correct adaptation at the level of intra- and extra-personal space. In the relationship with time, some instability was evident. However, he tended to show a greater ability to adjust to the individual and shared rhythm, especially in contact with the horse. Regarding the relationship with objects, in addition to using them in a concrete way, there was a progression in using them to communicate by attributing an affective and symbolic meaning to them. In the relationship with others (the therapist and horse), Tomás improved in the manifestation of his needs (verbally and non-verbally), with a progressive tendency in the type of relationship with the horse (tactile and, later, bodily), greater cooperation, and listening. Finally, the representations of Tomás that were initially disinvested and poorly detailed became more invested in graphic representations, with volume, details of clothing, and facial expressions often directed at the horse.

### 4.3. Case 3

#### Initial and Final Evaluation

Figure 7 reflects the overall impact of the intervention on Nuno’s competencies and difficulties from the perspectives of the parent and teacher.

This figure simultaneously presents the evolution of the total subscales at the time of pre- and post-psychomotor intervention as well as the total score of the difficulties. Nuno remained constant at the levels of behavior and symptomatic expression between the two evaluation moments.

In the school context, the total score of difficulties remained constant among the evaluations, obtaining a low value of 3 points. In both moments of the assessment, no behaviors of emotional expression were observed, and the subscale of behavior problems followed the same trend. As for the hyperactivity subscales and relationship problems with colleagues, the values obtained in both evaluations were reduced. At the level of the subscale of prosocial behaviors, the score of 6 in both evaluations was maintained, being within the average values. In the school context, no difficulties were identified on the part of Nuno, with a zero score obtained in all subscales except for prosocial behaviors, which initially scored 10 points and in the final evaluation scored 8 points, which was within a skewable and healthy value range. In conclusion, there were no significant oscillations between the evaluation moments, so a stable behavior was maintained, according to the respondents.

The results of the ACES are presented (Figure 8).

Figure 8a (subscale of facial expressions) reflects an initial difficulty in the identification of the facial expressions “scared”, “angry”, and “normal”, which improved after the intervention, with significant changes observed between the moments of evaluation. It is noteworthy that a maximum score was achieved in identifying the facial expression “sadness”. Figure 8b (subscale of situations) reflects that Nuno had difficulty identifying situations classified as “scared” and that he obtained maximum classifications in the identification of “happy” and “sad” situations in the second moment of evaluation. Regarding the behavior subscale, represented in Figure 8c, there was a tendency in the initial evaluation to incorrectly classify “scared” behaviors as “normal”. After the intervention, there was an improvement in the identification of emotions represented by behaviors, as it is possible to visualize in the second moment of evaluation.

Figure 9 allows an overview of Nuno’s performance, reflecting a significant evolution in correct emotional perception.

The data collected through the psychomotor observation sheet showed that, throughout the sessions, the Nuno revealed some instability in progressions in terms of movement. However, there was a tendency toward adequate body mobility. As for gesture, greater disinhibition was progressively observed, initially with the horse and at a later stage also with the therapist. There were no changes in the tone level, which was already adequate; however, there was an improvement in the level of horse riding. In the relationship with space, there was progressively greater exploration of it and a smaller invasion of the space of others, showing greater awareness of their personal space. Regarding the relationship with time, some instability is mirrored; however, there was a tendency toward a greater ability to adjust to the different times of the session and the adequacy of the individual and shared rhythm. As for the relationship with the objects, this relationship was initially used in a concrete way and evolved into creative use with the attribution of symbolic value. In the relationship with the others (the therapist and horse), there was a change in the type of relationship with the horse: initially more tactile and with a preference for nonverbal communication, and, at a later stage, a more bodily relationship involving the whole body. Highlighting the relationship with the horse, it became more open to communication, initiative, listening, and cooperation. In the relationship with the therapist, Nuno improved in the manifestation of needs (verbally and nonverbally), with an attitude of greater cooperation towards the game. Finally, graphic representations and constructions evolved, increasing in detail and often directed at the horse.

## 5. Discussion

Over four months, an intervention was performed (comprising 17 individual sessions, each lasting 45 min) in an equestrian context with three institutionalized children, aged between 5 and 7 years. In general terms, it was delineated with the common objectives of favoring the affective relationship, promoting relaxation, adhering to rules, solving problems with identification, and developing appropriate solutions, and developing a relationship with the horse.

### 5.1. Case 1

Regarding the evaluation of Maria’s abilities and behavioral difficulties, she showed: (i) a decrease in the scores of difficulties in both contexts, which was more pronounced in the domestic context (after the intervention, there was a decrease in behaviors with emotional expression); (ii) a significant reduction in the score of the hyperactivity subscale. The guardian did not observe difficulties in the subscale of behavior problems or problems in relationships with colleagues. This may be because Maria’s behaviors differ from those in the school context [29]. This was initially framed in the hyperactivity subscale and atypical values in the subscale of the scoring of difficulties in emotional behaviors, behavior problems, and relationship problems with colleagues. After the intervention, a positive evolution was observed in terms of hyperactivity with repercussions on the movement, which was more adjusted (in terms of space and time) in the equestrian context and gestuality. It is known that EAS contributes to greater body awareness, which may have contributed to this evolution [19]. Again, this positive trend of facial expressions and emotional identification in the relationship with colleagues became more appropriate in manifesting her needs more efficiently [28] and evolving from a tactile relationship with the horse to a body relationship. This situation indicates that improving interpersonal relationships has resulted in better emotional regulation and self-control. Moreover, the representations began to be more invested [28] and directed toward the horse, indicating an evolution in communication (verbal and nonverbal) [8].

### 5.2. Case 2

Initially, Tomás was at low risk of emotional and behavioral difficulties in domestic and academic contexts. After the intervention, he improved in this last context but regressed slightly in the first. At the end of the study, this improvement was shown to be associated with growth in emotional regulation and self-control because there was an improvement in his ability to associate situations with emotions and with these behaviors [8]. Of the more immediate consequences in the equestrian context, the most pronounced positive evolution was in the relationship with space and time and with others (the therapist and horse), resulting in greater body availability in the relationship with the horse as well as greater relational empathy, as suggested by Burgon [17].

The difficulty in identifying facial expressions of sadness that we associated with a slight behavioral regression was maintained throughout the study. It was believed that there had been few positive experiences with peers, resulting in less emotional adjustment at home [12].

### 5.3. Case 3

Nuno presented a low risk of emotional and behavioral difficulties before and after the intervention. Still, the intervention improved his emotional perception and gestures. This positive evolution resulted in greater body availability in the relationship with the horse, in the relationship with the therapist, in more intentional movements [28], and, finally, in a more explicit manifestation of his needs [8]. Given these developments, his representations also became more detailed and representative of the equestrian context [28].

### 5.4. General

The evaluation protocol used in this study was in line with what is mentioned above, which highlights that a large percentage of institutionalized children present behavior problems [3]. Two of the cases improved at this level with EAS. Nevertheless, the lack of agreement among the various respondents was evident in the presence of the same behaviors in a child. Stivanin et al. [29] warn that a child may exhibit different behaviors according to where they are, the environment around them, and the people they are related to, among other aspects. The teacher’s observation data are restricted to the classroom, where the child is exposed to rules and where behaviors are more controlled. At the same time, parents can identify behaviors at home, in public places, and in other contexts of the child [29].

As for the children’s ability to recognize emotions and the accuracy of their attribution, it is possible to state that overall and after the intervention, all children showed improvements in the identification level of facial expressions and improved emotional perception. Improving socio-emotional skills induces greater self-confidence in individuals, providing them with a greater capacity for emotional regulation and self-control [8]. Consequently, it contributes to a higher adaptation to the context [11,13] and a more significant emotional adjustment [12].

Facial, body, and empathic gestures were other parameters whose improvement was common to all participants. It is important to highlight that the empathic dimension is essential for healthy emotional and social functioning and that, in this study, it was possible to verify that this dimension was initially fragile [17,18]. The relationship between space, time, and objects also changed. The relationship with the others (the therapist and horse) revealed a positive progression in all participants, starting with a tactile relationship and evolving into a more bodily relationship (in which the whole body touches the horse). Consequently, children became more aware of the influence of their posture and behavior on the horse’s behavior. Regarding the representations and manifestations of needs, there was a significant evolution common to all, which is in line with the one advocated by Costa [28] and Domitrovich et al. [8]. The confrontation with new places, routines, and unknown people is sometimes challenging, mirrored through the difficulties in the relationships with space, time, objects, and others. Overcoming these difficulties and creating new affective and spatial references [6] associated with positive interactions contributes to improving children’s spatial and temporal organization [10]. These improvements were observed in children’s availability of some horsemanship skills (e.g., more planning in general horse care and horse leading).

The improvement in the participants’ socio-emotional competencies enabled the mastery of intrapersonal competencies, such as success at the level of self-regulation and self-control, allowing the development of positive thoughts and coping strategies [8].

It should be considered that EAS improved the socio-emotional skills of the three cases. It is believed that progression would have been more pronounced if there had been a higher frequency of sessions (e.g., biweekly). However, this was not possible due to the unavailability of the institution’s technicians and a constant change in the sessions and difficulties. In addition, the staff were not adequately trained to deal with the children, which could affect their psychic and body availability. Another aspect that may have influenced the results is the fact that there was negative interference from a fourth child, who did not meet the criteria for participation in this study, and who was in an exaggerated state of emotional lability, causing moments of enormous agitation for these children [12].

## 6. Conclusions

The study aimed to understand the effects of an EAS intervention on socio-emotional skills, contributing to an increase in knowledge in this domain, considering the reduced number of studies.

Through the three cases presented, a positive contribution was made in improving the socio-emotional skills of the participants, namely in favoring the affective relationship, adhering to rules, solving problems with identification and/or developing appropriate solutions, and developing a relationship with the horse. These had an impact on intrapersonal skills with marked improvement in self-regulation and self-control, in addition to an improvement in the intentionality of movement and adequacy of gesture to the context. These results gave children the possibility of developing new adaptation mechanisms. Moreover, improvements were observed in some horsemanship skills.

It is suggested that studies be conducted on a greater number of children in the future, which would include individual and group intervention with a higher frequency of sessions per week.

This type of intervention underlies a renewed educational and therapeutic approach, contributing to mental health promotion in this population.

## Figures and Tables

**Figure 1 ijerph-20-02846-f001:**
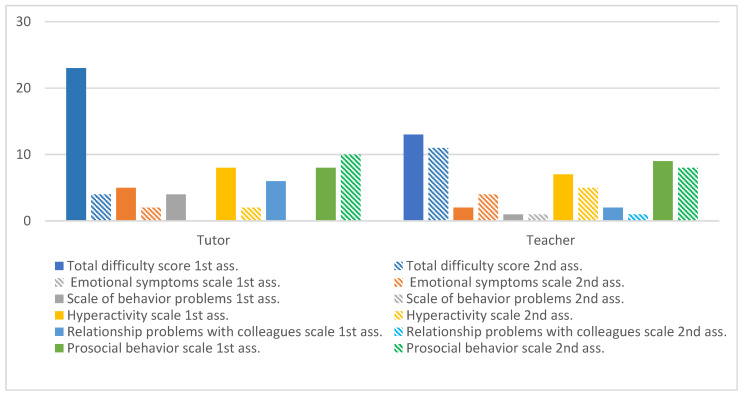
Maria’s global assessment of SDQ.

**Figure 2 ijerph-20-02846-f002:**
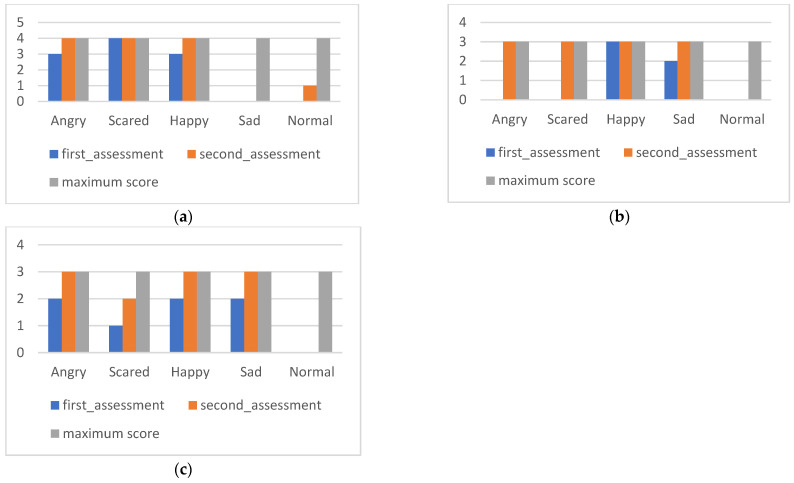
Maria’s ACES subscales: (**a**) facial expressions; (**b**) situations; (**c**) behaviors.

**Figure 3 ijerph-20-02846-f003:**
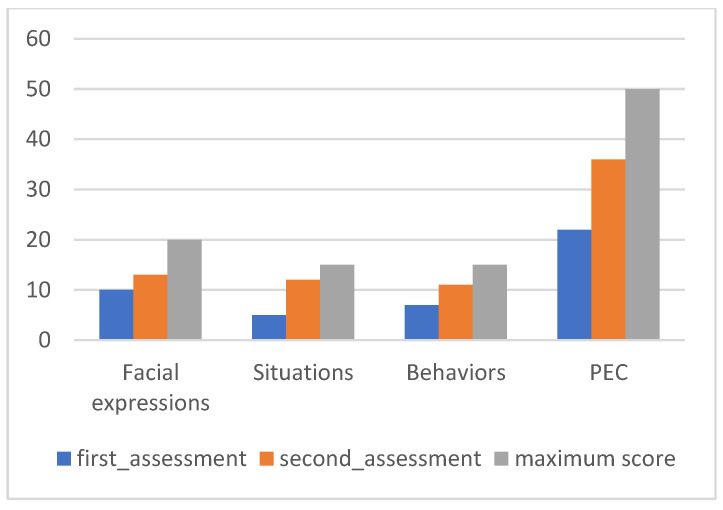
Maria’s PEC results.

**Figure 4 ijerph-20-02846-f004:**
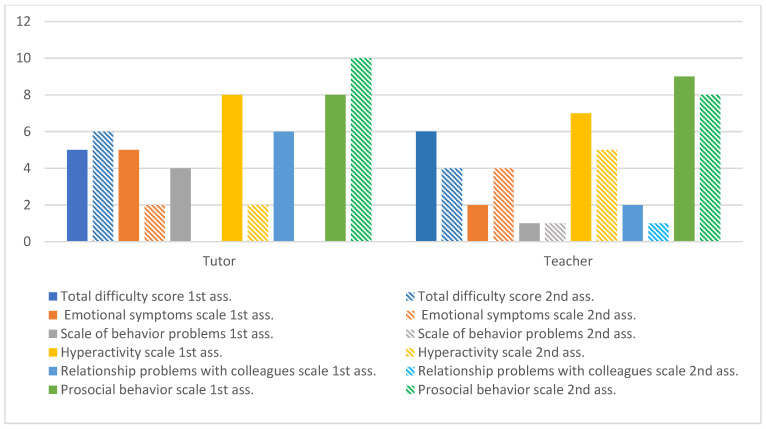
Tomás’s global assessment of SDQ.

**Figure 5 ijerph-20-02846-f005:**
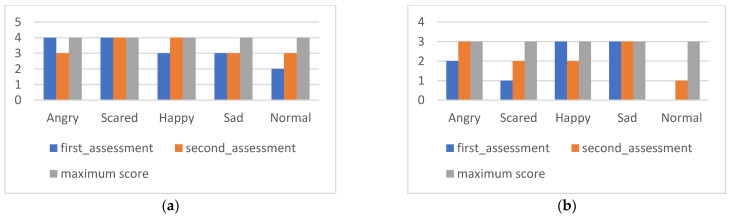
Tomás’s ACES subscales: (**a**) facial expressions; (**b**) situations; (**c**) behavior.

**Figure 6 ijerph-20-02846-f006:**
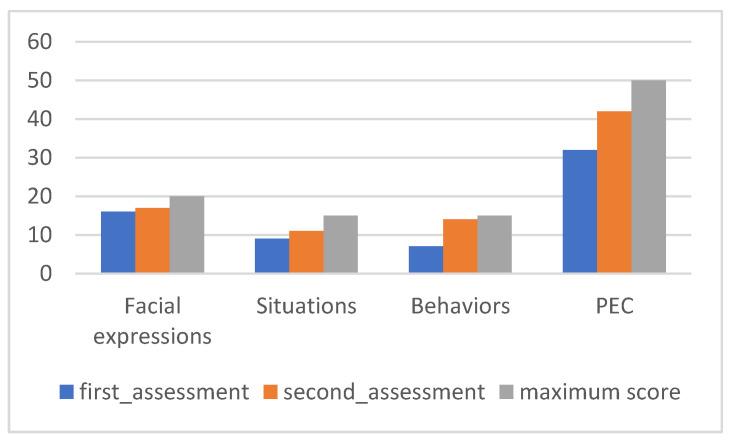
Tomás’s PEC results.

**Figure 7 ijerph-20-02846-f007:**
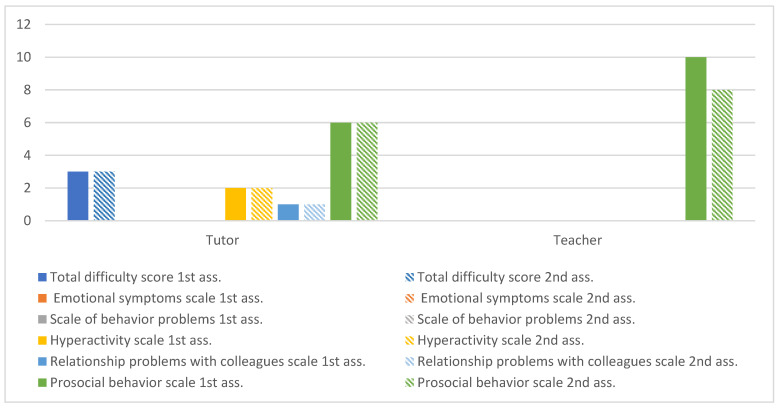
Nuno’s global assessment of SDQ.

**Figure 8 ijerph-20-02846-f008:**
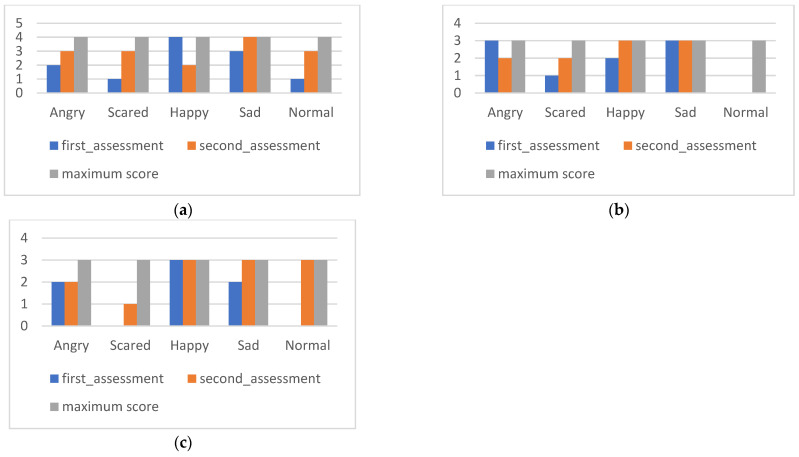
Nuno’s ACES subscales: (**a**) facial expressions; (**b**) situations; (**c**) behaviors.

**Figure 9 ijerph-20-02846-f009:**
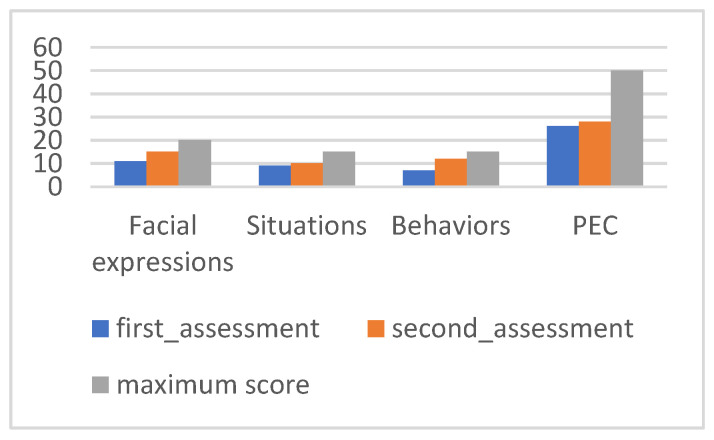
Nuno’s PEC results.

## Data Availability

The data presented in this study are available on request from the corresponding author. The data are not publicly available due to not disclosing the personal data of participants.

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
