# Peer review of "Equine-Assisted Therapeutic Intervention in Institutionalized Children: Case Studies"

_ijerph, 2023, doi:10.3390/ijerph20042846_

Round 1

Reviewer 1 Report

The authors did not sufficiently describe specifics of the intervention. Horse-human interactions and horse-assisted therapy can vary greatly in type of exercises. It is not standardized and needs to be detailed for the study. I could not provide a proper review without an adequate description of the HAT intervention. If this is corrected, that is proper description provided, then the manuscript would likely be worthy of publication and a probable contribution to the field. 

Author Response

Dear  Reviewer,

Kind regards,

Ana Rita Matias

Reviewer 2 Report

General

·         The paper is excessively long and would benefit from tight editing and shortening

·         The paper would benefit from English language editing.

Introduction

·         The introduction is overly long and not well focused.  It would benefit from shortening and editing.  However, it would also benefit from briefly reviewing more of the literature regarding the use of equine-assisted interventions for children

·         Should “psychic life” be “psychological life”?

·         What does “welcomed” mean in this context?

·         Should “(AAS)” be “(AAT)” animal-assisted therapy?

·         The authors state, “Within the AAs, we enhance Horse Assisted Therapy (HAT).”  What does ”enhance” mean in this context.

·         Horse Assisted Therapy (HAT) is not currently recommended terminology. See Wood W, Alm K, Benjamin J, et al. Optimal terminology for services in the United States that incorporate horses to benefit people: a consensus document. J Altern Complement Med. 2020; 27(1): 88–95.

Materials and Methods

·         Do you have diagnoses for the children?

·         Define “psychomotor approach” is this psychotherapy, hippotherapy or something else?

·         Provide more details about the intervention.  Was it mounted work, unmounted work, both?  Were the sessions all the same or different?

Results

·         A short summary paragraph including all the children would make understanding the results easier for readers

Discussion/conclusion

·         Be careful not to overstate results

·         How does this study contribute to gaps in the literature

·         Compare results to other studies of equine-assisted services for children

Author Response

(The authors gave the same response as above.)

Reviewer 3 Report

General comment: The manuscript “Horse-assisted therapeutic intervention in institutionalized children: case studies” is an interesting paper aiming to understand the effects of a HAT intervention with a psychomotor approach on socio-emotional competencies in institutionalized children. The interest of this project is based on the knowledge that when children are institutionalized the separation process can have a great impact on their psychic life, as well as in the behavioral disturbances, leading  to a loss of affective and spatial references. Then, it seems essential to understand whether this type of intervention can be useful as a renewed educational and therapeutic approach, contributing to the promotion of mental health in this population. The research project utilized a quantitative and qualitative assessment, by means of an Assessment of Children`s Emotion Scales (ACES), carried out on three children before and after the intervention, consisting in 17 sessions of HAT with a psychomotor approach, individually and weekly, and lasting approximately 45 minutes.  Considering the overall objectives of the study with these children, the data were collected by questionnaires and observational data, and the therapeutic goals, and respective therapeutic projects were designed based on the individual characteristics of each child, as well as a general intervention guide, seeking to encompass global needs. ACES represented an instrument consisting of three subscales (Facial Expressions (20 items that assess the ability to understand the emotions transmitted by the facial expressions of other children), Emotional situations (15 items that examine the child's ability to recognize emotional arousal in social situations), and Behaviors (15 items), that were applied in the context of an interview with the child. ACES aimed to evaluate the children's ability to recognize emotions and the accuracy of their attribution. The scale tended to evaluate the emotional knowledge of the estimated child through the ability that demonstrates to correctly associate with each stimulus (facial expression, situation, or behavior) one of five possible emotions (happy, sad, angry, frightened, normal). About the latter, the child is asked to identify the emotion presented by another child, among five possible response alternatives, contented, sad, angry, frightened, and finally normal. The result of this quotation in the three scales were presented as Correct Emotional Perception (PEC). Moreover, an individual psychomotor observation sheet was developed from the individual Observation Model described by Martinez et al. The utilized model is reported in Appendix 1 and it allowed to understand motor expression and gestuality, highlighting how the child used his body, his motor efficiency, the way he/she guided, how he related to objects and space, and others. With this form, movement, posture, tone, space, time, objects and relationship with others, and representation were studied. The results are then presented as case studies and precisely described and represented in graphs and figures.

The results reported an improvement in skills of children, with an impact on intrapersonal skills and with marked improvement in self-regulation and self-control, in addition to an improvement in the intentionality of movement and adequacy of gesture to the context.

The conclusions drawn suggested that these results gave children the possibility of developing new adaptation mechanisms. The Authors report a positive contribution in improving the socio-emotional skills of the participants, namely in favoring the affective relationship, adhering to rules, and solving problems with identification and/or development of appropriate solutions, development of the relationship with the horse.

These results could be a valuable tool in evaluating the effects of human-animal relationships in order to improve humans’ welfare. Then, the paper is interesting, because the contents addressed in this study are worthy of further investigations, from both the speculative and the applied points of view. However, although the beneficial effects of EAS on multiple aspects of physical and mental human needs are well assessed, the hypothesis of the study should require a deepest description of how the sessions of the therapeutic program were conducted to enforce the specific conclusions on the effects of EAS on children. From my point of view, this additional information, although generally presented, could be resumed in a more detailed way in the section “Materials and Methods”, and , may be, the qualitative changes could be also correlated to specific aspects of EAS improvements (either horsemanship skills such as proper brushing, leading, general horse care, psychomotor methodology or riding skills). Further information could add value to critical evaluation of the interesting results obtained, so to improve further speculative planning of research. Moreover, the existing literature background on the extent of stress really supported by trained therapy horses during activity should be also interesting to be considered. Nevertheless, the manuscript is attractive and easy to read and the results obtained are clearly presented.   

Title: It is suitable and it well describes the experiment presented in the manuscript.

Abstract: It is suitable. Abstract clearly identifies the interest for this research and its possible relevance. It recaps the information contained in the main text without repetitions.

Introduction: The Introduction provides adequate background. This section is concise, and includes some specific literature references. However, further literature references could be added.

Materials and Methods: The main concern of the paper regards this section, due to the lack of useful information on description of how the sessions of the EAS program were conducted. I suggest to add more information also about data concerning the horses involved in the study (e.g., training level, previous experience), and the specific therapeutic activities carried out. Age, in particular, and previous experience of therapeutic horses could largely influence the results. In the same time, the environmental conditions (presence of co-specific, presence of stranger people) during activity were not considered and further information could be valuable. No mentions are present concerning the characteristics of the therapeutic activities done by horses, which could significantly influence the results.

Results: The results are clear, although they could enjoy further value by considering the possible correlation with the more specific description of sessions carried out by the single patient. Results are presented clearly and logically, and the data are clearly presented in the graphs.

Discussion: The discussion is well organized and balanced. The comments reported in discussion are pertinent to the results achieved. The authors critically examine their results in the light of the state of science highlighted in the introduction. The discussion of results is extensive and clear.  Discussion follows a logical line, but the current knowledge on the topic could be more properly presented studying and adding further references about from the horses’ point of view. The conclusions are drawn from the study related to the aim of the study and potentially plausible in terms of the results obtained and applied in equine assisted services.  The interpretation of results proposed by the authors in the discussion could be shared. The paper offers the perspective for further study.

References: They are appropriate, although they could be reinforced by further items.

Tables and Figures: They are clear and explicative. I suggest to make a control and eventually reconsider the correspondence of the numbers of the graphs as they are reported in the text.

Decision: The current manuscript is acceptable for publication after minor revision.

Author Response

(The authors gave the same response as above.)
